# Personalized Layer Selection for Graph Neural Networks

**Kartik Sharma** *  *ksartik@gatech.edu*
*Georgia Institute of Technology*

**Vineeth Rakesh**  *vinmohan@visa.com*
*Visa Research*

**Yingtong Dou**  *yidou@visa.com*
*Visa Research*

**Srijan Kumar**  *srijan@gatech.edu*
*Georgia Institute of Technology*

**Mahashweta Das**  *mahdas@visa.com*
*Visa Research*

**Reviewed on OpenReview:** *https: // openreview. net/ forum? id= JyjTJAG9yZ*

## Abstract

Graph Neural Networks (GNNs) combine node attributes over a fixed granularity of the local graph structure around a node to predict its label. However, different nodes may relate to a node-level property with a different granularity of its local neighborhood, and using the same level of smoothing for all nodes can be detrimental to their classification. In this work, we challenge the common fact that a single GNN layer can classify all nodes of a graph by training GNNs with a distinct *personalized* layer for each node. Inspired by metric learning, we propose a novel algorithm, **MetSelect**, to select the optimal representation layer to classify each node. In particular, we identify a prototype representation of each class in a transformed GNN layer and then classify each node using the layer where the distance is smallest to a class prototype after normalizing with that layer's variance. Results on 10 datasets and 3 different GNNs show that we significantly improve the node classification accuracy of GNNs in a plug-and-play manner. We also find that using variable layers for prediction enables GNNs to be deeper and more robust to poisoning attacks. We hope this work can inspire future works to learn more adaptive and personalized graph representations.

## 1 Introduction

Graph Neural Networks (GNNs) (Hamilton et al., 2017; Kipf & Welling, 2016) show superior performance on various node-level and graph-level tasks for real-world applications, including fraud detection (Tang et al., 2022) and online content recommendation (Ying et al., 2018) . A GNN of depth $L$ iteratively learns $L$ different representations by combining a node's attributes with its local neighborhood Gilmer et al. (2017) and then use the final layer representation to classify each node. However, nodes in a graph differ from each other and their underlying property may correlate with a distinct granularity of each node's neighborhood. For instance, consider the task of predicting the political ideologies (*i.e.*, conservative or liberal) of users in a social network. Users tend to form echo chambers by interacting with people of similar ideologies (Cinelli et al., 2021), whereas some doubtful users may interact with diverse ideologies while leaning towards one (Boutyline & Willer, 2017). While the neighbor information may be useful to accurately classify the users within an echo chamber, it can confuse a classifier for the users that interact with both chambers where using just the

---

*Work done during internship at Visa Research

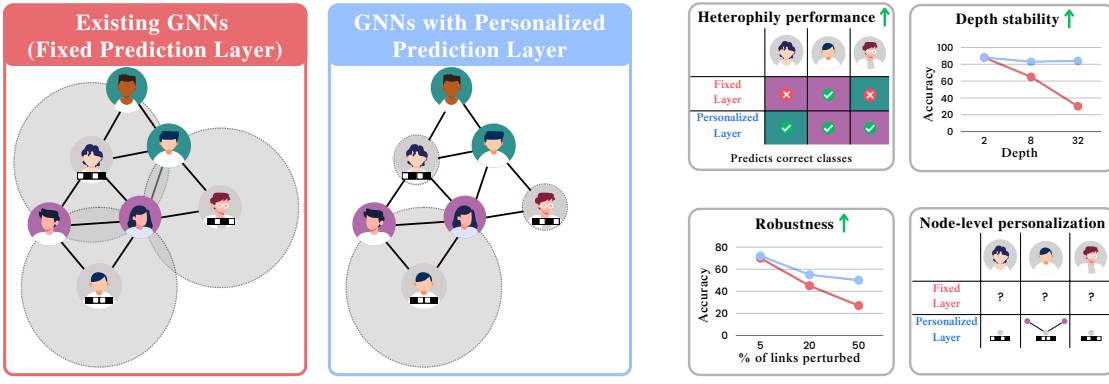

Figure 1: GNNs with personalized prediction layer can empower existing GNNs for more *accurate*, *personalized*, *robust*, and *deeper* node classification (*e.g.*, classifying unlabeled users (gray) as conservative (purple) or liberal (green)). For example, while some nodes benefit from their neighbors, others may be better off by predicting using only their attributes.

users' traits may be better. Figure 1 highlights the importance of using a personalized layer instead of a fixed layer of a GNN with this example.

We hypothesize that selecting a personalized GNN layer has wide-ranging implications. Existing GNNs are known to suffer from the problem of oversmoothing, where node representations become indistinguishable due to convolution (Nt & Maehara, 2019). This leads to poor performance on datasets where labels are typically associated with lower-level features. Existing works have focused on handling this issue by either enhancing the representations with higher frequency channels (Luan et al., 2022; Chien et al., 2020; Nt & Maehara, 2019; Eliasof et al., 2023; Zhao & Akoglu, 2019), continuous neural diffusion processes (Bodnar et al., 2022; Di Giovanni et al., 2022), or by rewiring the underlying graph (Rong et al., 2019; Gutteridge et al., 2023). However, we note that one can **enhance the accuracy** of existing GNNs by simply using a node-personalized representation for prediction instead of the final representation. We can also circumvent the problem of loss in performance in **deeper** GNNs (Li et al., 2018) by identifying the optimal layer for each node and thus, ignoring the node representations at higher depths if they don't help to classify that node. Using personalized GNN layer also boosts their **robustness** to untargeted training-time perturbations (Zügner et al., 2018; Sun et al., 2020; Xu et al., 2019). These perturbations are carefully crafted to reduce the overall accuracy of the GNNs over a given dataset but their effect will be limited since we can adaptively select the layer that is useful for classification and ignore any redundant information that may be perturbed. Finally, these improvements can be provided to any GNN in a **plug-and-play** manner while preserving their analytic simplicity by **personalizing** the selection of these representations.

In this work, we first formalize the problem of finding the optimal GNN layer for each node to solve the transductive node classification with a given GNN. Then, we propose a novel method, called **MetSelect** that leverages **Met**ric learning to **select** the optimal layer in an efficient manner. Here, we find a prototype representation of each class in transformations of each GNN layer and then compare the metric distance of a node to these prototypes across layers to find the optimal layer. To train the models, we minimize the cross entropy loss of the decoded probabilities from the optimal layers for each training node while minimizing their distance to the true class prototype in the transformed space. Through experiments on 10 datasets and 3 GNNs, we show that MetSelect boosts the node classification performance by up to 20%, especially on heterophilic datasets. We also find that MetSelect enables deeper GNNs by preserving the accuracies for as high as 10 layers. GNNs trained using MetSelect are also found to be robust against untargeted poisoning attacks (Zügner et al., 2018), giving up to 100% enhancement for different graph neural networks.

## 2    Related Work

**Layer aggregation.**    Layer aggregation methods such as JK-net, try to learn personalized representations by aggregating different representations through pooling operations and/or skip connections for node classification (Xu et al., 2018b; Dwivedi et al., 2023; Rusch et al., 2022). However, they do not adaptively select a single layer for classifying a given node but rather merge the representations for all nodes in the same way. This reduces the interpretability and increases the complexity of the underlying network by predicting through non-linear pooled transformation of the GNN representations. On the other hand, the simpler layer selection of MetSelect preserves the interpretability and the analytic simplicity of the underlying GNN as the predictions are still made through the GNN embeddings. Furthermore, layer aggregation methods require all representations to lie in the same space and to have the same dimension while selecting optimal layers through MetSelect completely relaxes such assumptions.

**Alternative GNN architectures.**    Based on the observations that a GCN can only approximate a low-pass filter (Nt & Maehara, 2019; Di Giovanni et al., 2022; Yan et al., 2022), new models have been proposed to address the heterophilic (Pei et al., 2020; Luan et al., 2022; Eliasof et al., 2023; Chien et al., 2020; Rossi et al., 2024) and closely-related oversmoothing (Bodnar et al., 2022; Yan et al., 2022) challenge. However, these models achieve these enhancements by modifying the underlying message-passing architecture, while we present a plug-and-play approach to show that these issues can possibly be addressed by adaptive layer selection. NDLS (Zhang et al., 2021) also finds node-specific smoothing level, their method is based on a simple feature smoothing model and is agnostic of the underlying GNN and the label distribution. Furthermore, while they hypothesize that the optimal layer is the one at which the influence of other nodes is similar to the stationary influence, we hypothesize that given a GNN, it is the one that minimizes the distance from the class prototype in the corresponding embedding space.

**Alternative training of GNNs.**    Alternative training strategies have been proposed to support larger graphs (Gandhi & Iyer, 2021), faster training (Cai et al., 2021), longer distances (Li et al., 2021; Gutteridge et al., 2023; Rong et al., 2019), robustness (Gosch et al., 2023), and a joint structure and embedding learning (Jin et al., 2020; Zeng et al., 2021). However, none of these approaches focus on our problem of selecting the optimal personalized layer and instead focus on increasing either robustness or faster convergence. Zeng et al. (2021) decouples the depth and the scope of the GNNs to consider a neighbor multiple times in the message passing by using a GNN of depth greater than the neighborhood scope of that node. On the other hand, our objective here is to find an optimal representation depth $L'$ strictly less than the full scope $L$ that best predicts the node's class.

**Neighbor importance.**    Certain methods aim to identify useful neighbors for each node separately to better classify them. Self-interpretable (Dai & Wang, 2021; Zhang et al., 2022; Feng et al., 2022) and attention-based (Veličković et al., 2017; Shi et al., 2020; Brody et al., 2021) graph neural networks weigh nodes based on the similarity between different nodes in the whole graph or the local graph structure. While this implicitly weighs each neighbor differently, the representations of all nodes are still formed from a fixed number of message-passing steps. Thus, these are still susceptible to oversmoothing and are not scalable as opposed to their graph neural network counterparts. Instead, we provide plug-and-play improvements to out-of-the-shelf GNNs by simply presenting a way to select the optimal smoothing level for each node.

**Metric Learning.**    Our strategy to choose the optimal layer is inspired by prior works in the image domain to learn embedding spaces by forming multiple clusters per class (Rippel et al., 2015; Deng et al., 2020). While it is prevalent to use distance-based classifiers in images (Deng et al., 2019; Salakhutdinov & Hinton, 2007) for learning robust embeddings, they have not been explored in the context of graph neural networks before. In this work, we use the embedding distance from class clusters to do personalized node-level smoothing.

## 3    Background

In this work, we study the problem of *transductive node classification*, where we are given a partially labeled graph and we want to learn the labels of the remaining nodes. More formally,

**Problem 1.** *(Transductive Node Classification) Given a graph* $\mathbf{G} = (\mathbf{X}, (\mathcal{V}, \mathcal{E}))$, *with node attributes* $\mathbf{X}$, *nodes* $\mathcal{V}$, *edges* $\mathcal{E}$, *and a set of labeled nodes* $\{(v, y) : v \in \mathcal{V}_{tr} \subset \mathcal{V}, y \in \mathcal{Y}\}$, *the objective is to learn a function* $f_{\mathbf{W}}(\cdot)$ *to predict the label* $y \in \mathcal{Y}$ *of each node* $v \in \mathcal{V}$.

We then consider an encoder-decoder architecture of $f_{\mathbf{W}}(\cdot) := \mathbf{g}_{\Phi} \circ \mathbf{h}_{\Theta}(\cdot) \in [0, 1]^{|\mathcal{Y}|}$, that predicts the probability of belonging to each class using a (trainable) decoder function $\mathbf{g}_{\Phi}$. The encoder $\mathbf{h}_{\Theta}$ is a message-passing Graph Neural Network (GNN) Kipf & Welling (2016); Gilmer et al. (2017) of depth $L$ that encodes each node $v_i$ as $\mathbf{h}_{\Theta}(v_i) := \mathbf{h}_i^{(L)} \in \mathbb{R}^d$ defined recursively as:

$$\mathbf{h}_i^{(l+1)} = \text{Update}(\mathbf{h}_i^{(l)}, \text{Agg}(\{\text{msg}(\mathbf{h}_j^{(l)}, \mathbf{h}_i^{(l)}) : (v_j, v_i) \in \mathcal{E}\})), \tag{1}$$

where $\text{Update}(\cdot), \text{Agg}(\cdot), \text{msg}(\cdot)$ are learnable functions, $\mathbf{h}^{(0)} = \mathbf{X}$, and the $l$-th **layer** $\mathbf{h}^{(l)}$ is a node representation formed using the neighbors at a distance of at most $l$ **hops** from that node, *i.e.*, $\mathcal{N}^{(l)}(v_i)$.

## 4 Problem

GNNs aggregate the $L$-hop information of each node to learn their label-separating representations. However, a major limitation stems from the fact that the same neighborhood scope $L$ is used to classify all the nodes. As shown in Figure 1, different nodes may require a different granularity of the local information to be classified properly, and providing more information could unnecessarily smooth out the classifying signal. To predict the user ideologies in a social network, nodes within an echo chamber may benefit from their 1-hop neighbors. However, a classifier may be better off ignoring the neighbors of the nodes that interact with both chambers and just using the attributes (0-hop). To capture such intraclass differences, we propose a novel problem that aims to classify a node using the optimal layer for each node. In other words,

**Problem 2.** *(Node-personalized layer selection in GNN) Given a graph* $\mathbf{G} = (\mathbf{X}, (\mathcal{V}, \mathcal{E}))$ *and a GNN encoder* $\mathbf{h}_{\Theta}(\cdot)$ *of depth* $L$, *the objective is to identify a personalized layer* $l^*(v) \in [0, L]$ *for each node* $v \in \mathcal{V}$ *such that* $\mathbf{h}^{(l^*(v))}(v)$ *best predicts the true label* $y(v)$.

As detailed in Section 1, this problem of selecting representations is important to study since it allows us to

1. **Increased accuracy** through personalized node-specific smoothing level.

2. **Enables deeper architectures** by disregarding over-smoothened representations.

3. **More robustness** against adversarial attacks as it ignores suboptimal layers.

4. **Plug-and-play** improvement irrespective of the underlying GNN architecture.

## 5 Methodology

While the original GNN learns a decoder $\mathbf{g}_{\Phi}(\cdot)$ over $\mathbf{h}^{(L)}$ to decode the labels, we compare the $L + 1$ representations formed by different layers of the encoder, *i.e.*, $\{\mathbf{h}_i^{(0)}, \mathbf{h}_i^{(1)}, \cdots, \mathbf{h}_i^{(L)}\}$ and pick the one most suitable to classify a given node. We call our method **MetSelect**[1], as we employ **Met**ric learning to **select** the node-optimal personalized layer to classify nodes. Figure 2 illustrates our method with an example of the transductive classification of a graph.

As noted in Equation 1, each layer of GNN $\mathbf{h}^{(l)}$ represents the nodes in distinct ways (typically, using the $l$-hop neighborhood). Thus, one can separate the nodes based on their labels in each of these spaces. Then, we propose to find the best layer $l^*(v)$ for a node based on how *confidently* that layer predicts its class. To compare the confidence of the prediction from different representation spaces, we adopt *distance*-based class decoding and use the distance as the notion of the confidence (Salakhutdinov & Hinton, 2007). However, distance-based decoding can distort the softmax decoding that we do to predict the labels from these layers. Thus, we propose to use a transformed representation $\tilde{\mathbf{h}}^{(l)} = \mathbf{W}\mathbf{h}^{(l)} + \mathbf{b}$ to find the best layer $l^*$.

---

[1]Code will be open-sourced after publication.

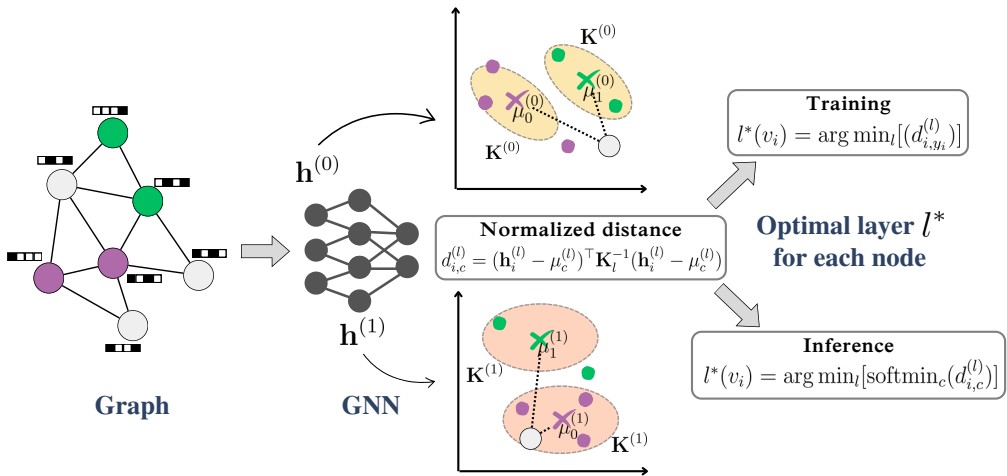

Figure 2: GNN with node-optimal prediction layer trained using MetSelect. Labeled nodes are denoted in purple and green while gray denotes unlabeled nodes.

Distance-based classifiers Rippel et al. (2015); Deng et al. (2019; 2020); Salakhutdinov & Hinton (2007) typically find a *prototype* embedding of each class and then classify the input to the class whose prototype is the closest in the embedding space. For instance, for layer $l$, the prototype of each class can simply be given as the **mean** embedding of the training nodes belonging to that class, *i.e.*,

$$\boldsymbol{\mu}_c^{(l)} = \frac{1}{\sum_{v_i \in \mathcal{V}_{tr}} \mathbb{1}[y_i=c]} \sum_{v_i \in \mathcal{V}_{tr}} \mathbb{1}[y_i = c]\tilde{\mathbf{h}}_i^{(l)}, \tag{2}$$

where $y_i$ is the label of the node $v_i$ and $\mathbb{1}[y_i = c] = 1$ if $y_i = c$ and 0 otherwise. However, we cannot simply compare the distances $\|\tilde{\mathbf{h}}_i^{(l)} - \boldsymbol{\mu}_{y_i}^{(l)}\|^2$ and $\|\tilde{\mathbf{h}}_i^{(l')} - \boldsymbol{\mu}_{y_i}^{(l')}\|^2$ where $l' \neq l$ since different spaces can have different class densities. Inspired by Rippel et al. (2015), we normalize the distance by the variance to discriminate densities in different spaces. In particular, we find the **variance** for each space $\tilde{\mathbf{h}}^{(l)}$ as simply the empirical variance given the means,

$$\mathbf{K}_l = \tfrac{1}{|\mathcal{V}_{tr}|-1} \sum_{v_i \in \mathcal{V}_{tr}} (\tilde{\mathbf{h}}_i^{(l)} - \boldsymbol{\mu}_{y_i}^{(l)})(\tilde{\mathbf{h}}_i^{(l)} - \boldsymbol{\mu}_{y_i}^{(l)})^\top = \tfrac{1}{|\mathcal{V}_{tr}|-1} \sum_{v_i \in \mathcal{V}_{tr}} (\tilde{\mathbf{h}}_i^{(l)})(\tilde{\mathbf{h}}_i^{(l)})^\top - \sum_c \mathbb{1}[y_i = c](\boldsymbol{\mu}_c^{(l)})(\boldsymbol{\mu}_c^{(l)})^\top. \tag{3}$$

Then, we propose **MetSelect** that selects the personalized layer $l^*(v)$ such that it minimizes the **normalized distance** or the Mahlanabois distance, $d_{i,c}^{(l)} = (\tilde{\mathbf{h}}_i^{(l)} - \boldsymbol{\mu}_c^{(l)})^\top \mathbf{K}_l^{-1}(\tilde{\mathbf{h}}_i^{(l)} - \boldsymbol{\mu}_c^{(l)})$. In particular,

$$l^*(v_i) = \arg\min_{l \in [0,L]} \min_{y \in [1,|\mathcal{Y}|]} \left[ \exp(-d_{i,y}^{(l)}) / \sum_{c=1}^{|\mathcal{Y}|} \exp(-d_{i,c}^{(l)}) \right]. \tag{4}$$

For training, since we know the true label of each node in the training set, we find the optimal layer as

$$l^*(v_i|y_i) = \arg\min_{l \in [0,L]} \left[ \exp(-d_{i,y_i}^{(l)}) / \sum_{c=1}^{|\mathcal{Y}|} \exp(-d_{i,c}^{(l)}) \right]. \tag{5}$$

**Training.** Existing GNNs decode the labels from final layer representations by training an additional neural network $\mathbf{g}_\Phi$ to minimize $\mathcal{L}_{CE}(v_i, y_i) = \log(\mathbf{g}_\Phi \circ \mathbf{h}_i^{(L)})_{y_i}$, where $y_i$ is the actual label of $v_i$. Since we want to decode from different layers, we employ $L + 1$ different decoders $\mathbf{g}_\Phi^{(l)}$ for $l \in [0, L]$ specialized for each layer. Thus, we train this using a *personalized* Cross-Entropy loss as

$$\mathcal{L}_{CE}(v_i, y_i) = - \log \big(\mathbf{g}_\Phi^{(l^*(v_i|y_i))} \circ \mathbf{h}_i^{(l^*(v_i|y_i))}\big)_{y_i}. \tag{6}$$

Since learning one decoder per layer can be hard especially for higher number of layers, we also propose to directly leverage the class moments as a distance-based classification. Here, we minimize the distance of a training node from the prototype of the correct class in the closest layer $l$ and maximize the distance from the prototypes of other classes in all layers. Assuming $\alpha = 1$, we propose

$$\mathcal{L}_{\text{distance}}(v_i, y_i) = \max(0, \alpha + d_{i,y_i}^{(l^*(v_i|y_i))} + \log \sum_{l=0}^{L} \sum_{j \neq y_i} \exp(-d_{i,j}^{(l)})), \tag{7}$$

However, we only use $\mathcal{L}_{\text{CE}}$ in the experiments unless otherwise mentioned.

Algorithm 1 describes the training steps in more detail. In particular, we first find the class means and variance for each layer and then use these values to minimize the loss $\mathcal{L}$ for each training pair $(v_i, y_i) : v_i \in \mathcal{V}_{tr}$. Also, note that we treat the two moments $\boldsymbol{\mu}_c^{(l)}, \mathbf{K}_l$ as constants in the loss function so that no gradients pass through them for the parameters $\mathbf{W}$ to simplify the gradient descent.

**Time Complexity.** The total training time taken per epoch by Algorithm 1 includes the time for (1) Finding the moments of each class cluster, (2) Finding the optimal layer for each node, (3) Calculating the loss over the training nodes, and (4) Updating the parameters using gradient descent. One can find the means $\boldsymbol{\mu}_c^{(l)}$ in time $O(L|\mathcal{Y}||\mathcal{V}_{tr}|d)$ and the variances $\mathbf{K}_l$ in time $O(L|\mathcal{V}_{tr}|d^2)$, where $d$ denotes the embedding dimension. The optimal layers can be found using $O(|\mathcal{V}_{tr}|L|\mathcal{Y}|d^2)$. One can then decode the probabilities and accumulate the loss in time $O(|\mathcal{V}_{tr}||\mathcal{Y}|d)$. This makes up $O(|\mathcal{V}_{tr}|L|\mathcal{Y}|d^2))$ while the forward pass through the GNN takes at least $O(Ld^2)$ (due to multiplication with $d \times d$ weight matrices). Thus, MetSelect remains efficient as long as $|\mathcal{V}_{tr}|$ and $|\mathcal{Y}|$ are small.

Algorithm 1: Training MetSelect

**Require:** GNN $\mathbf{h}$ of depth $L$, Training nodes $\mathcal{V}_{tr} \subset \mathcal{V}$, Number of epochs $N_e$, Learning rate $\eta$
1: $N_c \leftarrow \sum_i \mathbb{1}[y_i = c]$ for all classes $c$.
2: **for** $e = 1$ to $N_e$ **do**
3:     Initialize $\mathcal{L} \leftarrow 0, \widehat{\boldsymbol{\mu}}_c^{(l)} \leftarrow \mathbf{0}, \widehat{\mathbf{K}}_{l,0} \leftarrow \mathbf{0}$.
4:     **for** $v_i \in \mathcal{V}_{tr}$ **do**
5:         **for** $l \in [0, L]$ **do**
6:             $\widehat{\boldsymbol{\mu}}_{y_i}^{(l)} \leftarrow \widehat{\boldsymbol{\mu}}_{y_i}^{(l)} + \mathbf{h}_i^{(l)}$.
7:             $\widehat{\mathbf{K}}_{l,0} \leftarrow \widehat{\mathbf{K}}_{l,0} + \mathbf{h}_i^{(l)}(\mathbf{h}_i^{(l)})^\top$.
8:         **end for**
9:         $\mathcal{L} \leftarrow \mathcal{L} + \mathcal{L}(v_i, y_i; l^*)$ [Eqns. 5, 6].
10:     **end for**
11:     $[\boldsymbol{\Theta}, \boldsymbol{\Phi}, \mathbf{W}] \leftarrow [\boldsymbol{\Theta}, \boldsymbol{\Phi}, \mathbf{W}] - \eta \nabla \mathcal{L}$.
12:     $\boldsymbol{\mu}_c^{(l)} \leftarrow \widehat{\boldsymbol{\mu}}_c^{(l)}/N_c$.
13:     $\mathbf{K}_l \leftarrow \frac{1}{|\mathcal{V}_{tr}|-1}(\widehat{\mathbf{K}}_{l,0} - \sum_c N_c \boldsymbol{\mu}_c^{(l)}(\boldsymbol{\mu}_c^{(l)})^\top)$.
14: **end for**

While we empirically find the running time to be similar, specific sampling techniques to reduce the graph size can also be used to scale to larger graphs (Chiang et al., 2019).

**Space Complexity.** As shown in Algorithm 1, we calculate the means and variance by updating them in an online manner over each node. Thus, the space overhead of MetSelect does not depend on the number of nodes as it only involves $O(L|\mathcal{Y}|d)$ to store the means and $O(Ld^2)$ to store the variance.

## 6 Experimental Setup

**Datasets.** We consider 4 standard homophilic co-citation network datasets — Cora, Citeseer, Pubmed (Kipf & Welling, 2016) and ogbn-arxiv (ogba) (Hu et al., 2020), where each node represents a paper that is classified based on its topic area. We also used 6 heterophilic datasets — Actor, Chameleon, Squirrel, Cornell, Wisconsin, Texas (Pei et al., 2020). Following Pei et al. (2020), we evaluate the models on 10 different random train-val-test splits for all the datasets except ogba, where we used the standard OGB split. Table 1 notes the statistics of these datasets along with label homophily, which measures the proportion of nodes with the same labels with an edge in common.

Table 1: Statistics of the datasets used in this work. Label Homophily measures the proportion of nodes with same labels that share an edge.

| Dataset | # Nodes $|\mathcal{V}|$ | # Edges $|\mathcal{E}|$ | # Features $|\mathbf{X}|$ | # Labels $|\mathcal{Y}|$ | Label Homophily |
|---|---|---|---|---|---|
| Cora | 2708 | 5429 | 1433 | 7 | 0.83 |
| Citeseer | 3327 | 4732 | 3703 | 6 | 0.71 |
| Pubmed | 19717 | 44338 | 500 | 3 | 0.79 |
| Actor | 7600 | 33544 | 931 | 5 | 0.24 |
| Chameleon | 2277 | 36101 | 2325 | 5 | 0.25 |
| Squirrel | 5201 | 217073 | 2089 | 5 | 0.22 |
| Cornell | 183 | 295 | 1703 | 5 | 0.11 |
| Wisconsin | 251 | 499 | 1703 | 5 | 0.16 |
| Texas | 183 | 309 | 1703 | 5 | 0.06 |
| Ogbn-arxiv | 169343 | 1166243 | 128 | 40 | 0.65 |

**Graph Neural Networks.** We use 3 representative base GNNs [2] to assess the plug-and-play improvement of our method — Graph Convolutional Network (GCN) (Kipf & Welling, 2016), Graph Attention Network (GAT) (Veličković et al., 2017), and Graph Isomorphism Network (GIN) (Xu et al., 2018a). All models have depth $L = 2$ unless otherwise mentioned.

**Baselines.** While no baseline exists that directly finds the optimal layer for each node, we extend existing methods for our task to compare them. We use GNN+{name} to denote method "name" and just GNN to denote GNN trained with FinalSelect.

---

[2] https://pytorch-geometric.readthedocs.io/en/latest/modules/nn.html

- FinalSelect: Here, we simply use existing GNN implementations that consider the final layer representation for all nodes to predict their labels.

- NDLSelect (Zhang et al., 2021): NDLS is a node-dependent local smoothing method that trains a simplified GCN (Wu et al., 2019) with varying smoothing levels for each node. We use these node-specific layers $l(v_i)$ to train GNNs using Equation 6 and classify each node accordingly. Note that these personalized layers are agnostic of models and label distribution.

- AttnSelect (Xu et al., 2018a): We replace the softmax in JKNet-LSTM to max such that it selects a fixed layer for each node and train using Equation 6 for these node-specific layers.

**Implementation.** In GNNs, we assume $\mathbf{h}^{(0)} = \mathbf{X}$ but the feature dimension can be very high. Since in MetSelect, we find distances in the embedding space $\mathbf{h}^{(l)}$, a high dimension is not suitable due to the curse of dimensionality (Jordan & Mitchell, 2015). Thus, we used a trainable linear layer to transform the features into a $d$-dimensional space, *i.e.*, $\mathbf{h}_i^{(0)} = \mathbf{W}_0\mathbf{X}_i + \mathbf{b}_0 \in \mathbb{R}^d$. We also found other dimensionality reduction methods such as PCA and random transformation to perform similarly. All the models were trained using an Adam optimizer for 500 epochs with the initial learning rate tuned between $\{0.01, 0.001\}$. The best-trained model was chosen using the validation accuracy and in the case of multiple splits, the mean validation accuracy across splits. All the experiments were conducted on Python 3.8.12 on a Ubuntu 18.04 PC with an Intel(R) Xeon(R) CPU E5-2698 v4 @ 2.20GHz processor, 512 GB RAM, and Tesla V100-SXM2 32 GB GPUs.

**Metrics.** Following existing works, we use the **micro-F1** or **accuracy** score across different classes for the test set (Test ACC), validation set (Val ACC), and train set (Train ACC) to assess the performance. We found that the trends are similar for other aggregations of F1 across different classes such as class-weighted weighted-F1 and macro-F1 scores.

## 7    Results

Here, we show empirically that while solving Problem 2, one can make GNNs more *accurate*, *deeper*, and *robust* in a *plug-and-play* manner.

### 7.1    Can MetSelect be used to enhance the performance of GNNs in a *plug-and-play* manner?

Here, we test the impact of using a personalized prediction layer to classify the nodes from different GNN representations. Table 2 reports the performance of different layer selection methods on different GNNs and datasets. One can note that our proposed MetSelect method almost always gives the best accuracy among the baselines. This can be observed from the lowest average difference from the maximum accuracy ($\leq 0.015$). Standard GNNs that only use the final layer to classify all nodes are known to obtain good performance in homophilic datasets (Hu et al., 2020). In these cases, the three personalized layer selection methods (NDLSelect, AttnSelect, and MetSelect) preserve the performance of the test set across different GNNs for these datasets, with MetSelect being the most consistent among the three.

GNNs trained using MetSelect can ignore a node's neighbors at a scope that is not useful for its classification. This explains the overall boost in performance on heterophilic datasets. Table 2 shows that MetSelect can indeed improve/match the performance of the GNNs with a positive increase in the average performance for GCN, GAT, and GIN. In particular, we find a constant improvement in the extremely label-heterophilic datasets such as Cornell, Texas, and Wisconsin. This is particularly because MetSelect learns to ignore the graph structure when predicting labels for many nodes since the graph structure is not aligned with the label distribution for these datasets. Note that this is identified by MetSelect in an automated manner without specifically encoding this preference in the method anywhere. We also note that performance on homophilic datasets does not show a significant increase, which can be explained due to the fact that MetSelect does not enhance the representation power of these models and selecting the most representative final layer tends to be the most powerful for these datasets. Overall, we find that MetSelect provides the best overall performance, being at least 4 times closer to the best performance across datasets.

Table 2: Test F1-score of the GNN models averaged over different splits (except Ogba) in the datasets with standard deviation in the subscript. MetSelect-max inverts the proposed MetSelect method by selecting the layer that maximizes the distance from the class prototype. $\Delta$ Max denotes the average difference from the mean maximum value across datasets.

| | | Cora | Citseer | Pubmed | Actor | Chameleon | Squirrel | Cornell | Wisconsin | Texas | Ogba | $\Delta$ Max $\downarrow$ |
|---|---|---|---|---|---|---|---|---|---|---|---|---|
| GCN | FinalSelect | $84.99_{1.43}$ | $74.46_{1.72}$ | $87.11_{0.41}$ | $30.40_{1.01}$ | $68.36_{1.85}$ | $52.78_{1.34}$ | $58.65_{8.26}$ | $61.57_{6.68}$ | $67.03_{7.62}$ | $67.75$ | $2.77$ |
| | NDLSelect | $85.15_{1.38}$ | $75.98_{1.70}$ | $86.57_{0.41}$ | $28.53_{1.51}$ | $63.75_{1.96}$ | $51.14_{1.52}$ | $43.24_{6.62}$ | $50.98_{7.10}$ | $58.38_{6.14}$ | $63.35$ | $7.37$ |
| | AttnSelect | $84.87_{1.13}$ | $73.23_{1.32}$ | $86.28_{0.81}$ | $29.68_{1.73}$ | $58.20_{4.96}$ | $31.35_{2.22}$ | $53.78_{11.71}$ | $61.18_{6.84}$ | $61.89_{7.03}$ | $67.29$ | $7.31$ |
| | MetSelect (ours) | $84.23_{1.53}$ | $74.44_{1.51}$ | $86.60_{0.44}$ | $30.10_{0.79}$ | $66.28_{2.57}$ | $53.66_{1.74}$ | $68.92_{6.40}$ | $76.47_{6.98}$ | $64.05_{7.86}$ | $66.35$ | **$0.97$** |
| | MetSelect-max | $76.74_{2.61}$ | $70.48_{2.13}$ | $84.45_{0.60}$ | $29.39_{1.44}$ | $53.63_{1.70}$ | $37.70_{1.53}$ | $50.27_{10.54}$ | $58.63_{4.54}$ | $66.35_{5.30}$ | $55.28$ | $9.79$ |
| GAT | FinalSelect | $84.61_{1.72}$ | $74.23_{1.65}$ | $86.68_{0.54}$ | $29.13_{0.94}$ | $64.69_{1.63}$ | $46.75_{1.76}$ | $54.05_{6.37}$ | $60.00_{6.28}$ | $62.43_{7.80}$ | $68.58$ | $5.83$ |
| | NDLSelect | $83.48_{1.43}$ | $74.61_{1.65}$ | $85.69_{0.58}$ | $28.68_{1.34}$ | $62.39_{2.42}$ | $44.84_{1.66}$ | $42.16_{7.00}$ | $53.92_{6.81}$ | $60.27_{3.61}$ | $65.58$ | $8.79$ |
| | AttnSelect | $71.25_{1.94}$ | $73.99_{1.35}$ | $85.75_{0.42}$ | $32.30_{2.17}$ | $49.71_{2.41}$ | $36.69_{2.65}$ | $70.54_{3.24}$ | $80.78_{4.22}$ | $77.03_{2.92}$ | $58.84$ | $5.26$ |
| | MetSelect (ours) | $82.80_{1.60}$ | $74.53_{1.93}$ | $85.94_{0.43}$ | $28.99_{1.21}$ | $62.27_{2.59}$ | $48.49_{2.40}$ | $70.27_{4.23}$ | $81.96_{4.60}$ | $73.24_{7.15}$ | $66.25$ | **$1.47$** |
| | MetSelect-max | $80.41_{2.29}$ | $70.16_{2.16}$ | $84.72_{0.60}$ | $29.76_{1.20}$ | $57.37_{2.94}$ | $44.60_{1.99}$ | $46.22_{7.69}$ | $53.53_{6.83}$ | $61.22_{4.90}$ | $58.26$ | $10.32$ |
| GIN | FinalSelect | $81.63_{1.56}$ | $69.55_{2.57}$ | $87.19_{0.53}$ | $27.64_{1.22}$ | $68.95_{2.75}$ | $49.88_{2.83}$ | $51.08_{8.59}$ | $61.18_{5.13}$ | $65.14_{8.59}$ | $68.37$ | $5.62$ |
| | NDLSelect | $80.72_{1.52}$ | $71.79_{1.63}$ | $86.36_{0.50}$ | $26.55_{1.05}$ | $67.26_{2.41}$ | $53.46_{2.27}$ | $50.27_{8.57}$ | $54.12_{5.48}$ | $60.54_{6.14}$ | $66.18$ | $6.95$ |
| | AttnSelect | $81.29_{1.62}$ | $71.64_{2.17}$ | $86.57_{0.54}$ | $27.62_{1.96}$ | $61.36_{4.00}$ | $41.58_{4.48}$ | $60.00_{10.80}$ | $66.08_{6.06}$ | $70.27_{6.62}$ | $64.51$ | $5.58$ |
| | MetSelect (ours) | $80.97_{1.52}$ | $72.65_{2.70}$ | $86.73_{0.43}$ | $30.36_{1.21}$ | $63.29_{2.75}$ | $48.56_{2.49}$ | $69.46_{4.04}$ | $79.02_{3.07}$ | $75.68_{4.41}$ | $66.76$ | **$1.33$** |
| | MetSelect-max | $76.29_{1.81}$ | $68.97_{2.43}$ | $85.01_{0.63}$ | $27.59_{1.36}$ | $56.13_{2.36}$ | $41.37_{1.82}$ | $49.86_{9.50}$ | $59.12_{4.73}$ | $62.97_{5.20}$ | $55.83$ | $10.36$ |

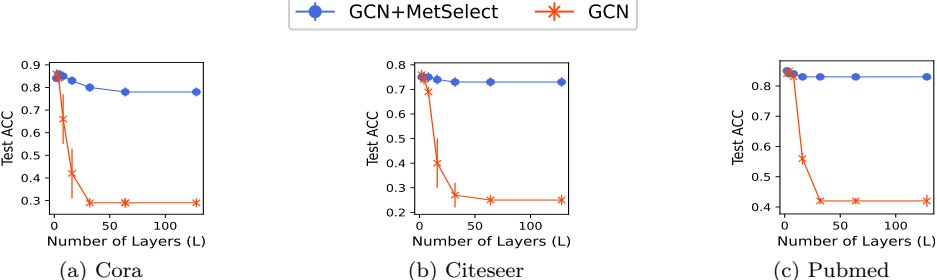

(a) Cora     (b) Citeseer     (c) Pubmed

Figure 3: Test Accuracy of GCN and GCN+MetSelect with varying depths for different datasets.

## 7.2 Does MetSelect enable *deeper* GCN models?

To test MetSelect's effect at different depths, we use the distance-based loss function in Equation 6 to efficiently train the model at larger depths but find similar results to hold for the linear loss (*i.e.*, Equation 6) till depth $L = 8$. Since we optimally choose the prediction layer to classify each node, we expect the performance of GCN+MetSelect models to preserve the test performance even when we increase the number of GNN layers. Figure 3 demonstrates this effect on Cora, Citeseer, and Pubmed datasets. We find that the accuracy of GCN+MetSelect drops by only up to 10 accuracy points even for a depth as high as 128, while the standard GCN model drops by at least 50 points with just 32 layers. This shows that MetSelect can indeed enable much deeper GCNs, thus increasing its representative power. For ogbn-arxiv, we found that while GCN's accuracy reduces from 0.67 to 0.22 and 0.06 at a depth of 16 and 32, while MetSelect preserves the accuracy with a slight reduction from 0.66 to 0.65 and 0.62.

## 7.3 Are the GNNs trained using MetSelect more *robust* to poisoning attacks?

Untargeted poisoning attacks (Zügner et al., 2018; Sun et al., 2020) have demonstrated that the mean GNN performance can suffer heavily from a change in the graph structure during training. In Section 4, we have highlighted that MetSelect can potentially make the models more robust to such untargeted attacks by ignoring the neighborhood scope not useful for a node's prediction and, thus, potentially the perturbations. To test this, we use Mettack (Zügner et al., 2018) to perturb our training graph that is used to train a GCN model using the standard and the proposed MetSelect loss. In particular, we use the black-box version of Mettack, which finds the attacks using a surrogate GCN model trained on the same dataset. These attacks are structural and represent the flip of a link, *i.e.*, add the link if it doesn't exist and delete it otherwise. The

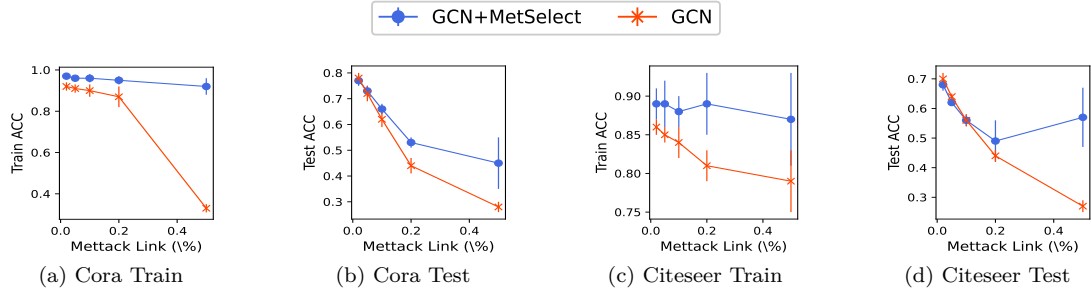

Figure 4: Accuracy of GCN and GCN+MetSelect after different strengths of Mettack perturbations.

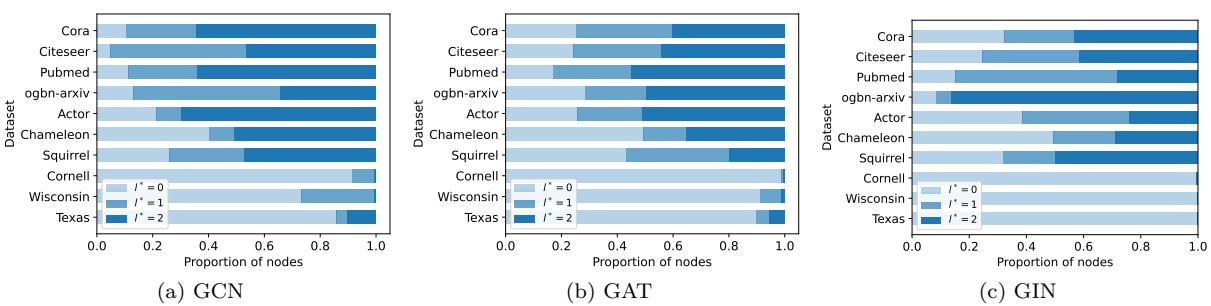

Figure 5: Proportion of test nodes with a particular personalized layer $\ell^* \in [0, 1, 2]$ as identified by MetSelect.

total number of links flipped, or the *budget* is fixed as a parameter that we set to be $p \times |\mathcal{E}|$ where we vary $p \in [0, 0.5]$ denoted by Mettack Link (%).

Figure 4 shows the robustness of the GCN model for Cora and Citeseer at different budgets using just the final layer (denoted as just GCN) and using personalized layer trained using Equation 7 (denoted as GCN+MetSelect). We find that the GCN trained using MetSelect is significantly more robust than the standard GCN with the training accuracy of GCN+MetSelect remaining unaffected by the attacks introduced during training. For instance, while the final layer fails to learn an accurate classifier even for the training set at higher budgets of 50%, MetSelect is able to still learn the representations to classify the training set. This is also reflected in the improved generalization of these classifiers as the test accuracy of the GCN+MetSelect is also significantly higher than GCN after the attacks. In particular, we see an improvement of up to 30 points in the test accuracy of GCN trained using MetSelect over the standard GCN. This establishes the robustness of the MetSelect loss owing to its variable prediction layer for nodes.

### 7.4 Analysis on MetSelect

**Ablation.** Here, we assess how much the node classification performance depends on the prediction layer identified by MetSelect. To quantify this, we compare it with a method that picks the polar opposite layer of what MetSelect would select. In particular, we consider the one that maximizes the normalized distance from the class center. Table 2 shows the effectiveness of $l^*(v)$ chosen by the MetSelect model. One can observe that MetSelect almost always outperforms the MetSelect-max (only exception being a little reduction in Actor for GAT) with an average positive gain in accuracy of over 9 F1 points.

**Self-loops.** We further study the effect of including the feature information in the GNN and see if the gains remain consistent even in this setting. Particularly, we include the node features by adding a self-loop on each node which ensures that the node information is accounted for in all the aggregation layers (Hamilton et al., 2017). Table 4 shows that the performance remains consistently higher than the final layer selection baseline while highlighting the plug-and-play benefits of our method.

Table 4: Comparison of FinalSelect and MetSelect for GCN with self-loops.

| | Cora | Citseer | Pubmed | Actor | Chameleon | Squirrel | Cornell | Wisconsin | Texas | Ogba | $\Delta$ Max $\downarrow$ |
|---|---|---|---|---|---|---|---|---|---|---|---|
| FinalSelect | $86.18_{1.15}$ | $74.71_{1.66}$ | $87.02_{0.45}$ | $30.80_{0.92}$ | $68.62_{1.81}$ | $55.99_{1.02}$ | $58.11_{9.56}$ | $61.96_{7.23}$ | $67.84_{5.76}$ | $69.72$ | $2.68$ |
| MetSelect | $83.72_{1.87}$ | $73.05_{1.69}$ | $86.03_{0.56}$ | $30.99_{1.24}$ | $63.75_{2.68}$ | $51.22_{2.47}$ | $66.22_{6.53}$ | $76.67_{5.10}$ | $71.62_{8.18}$ | $66.77$ | $\mathbf{1.77}$ |

Table 5: Comparison of MetSelect and layer aggregation methods in GCN using F1/100.

| | Cora | Citseer | Pubmed | Actor | Chameleon | Squirrel | Cornell | Wisconsin | Texas | Ogba | $\Delta$ Max $\downarrow$ |
|---|---|---|---|---|---|---|---|---|---|---|---|
| JKNet-Max | 0.86 | 0.76 | 0.88 | 0.36 | 0.62 | 0.40 | 0.62 | 0.75 | 0.67 | 0.68 | 0.023 |
| JKNet-Cat | 0.86 | 0.76 | 0.88 | 0.35 | 0.61 | 0.42 | 0.66 | 0.78 | 0.71 | 0.69 | 0.011 |
| JKNet-LSTM | 0.71 | 0.72 | 0.88 | 0.35 | 0.50 | 0.29 | 0.70 | 0.79 | 0.75 | 0.68 | 0.046 |
| JKNet-MetSelect | 0.84 | 0.74 | 0.88 | 0.36 | 0.58 | 0.35 | 0.67 | 0.79 | 0.75 | 0.67 | **0.009** |

**Running Time.** We theoretically found the running time complexity of MetSelect to be higher than existing GNNs. However, we find empirically that it remains within an acceptable factor of $< 2.5$ across different datasets to the original method. Table 3 shows the mean per-epoch time on the largest dataset (ogbn-arxiv) and the remaining small datasets. As compared to NDLSelect, we remain within reasonable bounds of $< 1.5$, while sometimes doing even better. Thus, our method is equivalent to the one-time cost of training a proxy model as is done in NDLSelect.

Table 3: Mean time taken by different algorithms for Ogba and other small datasets.

| | | FinalSelect | NDLSelect | MetSelect |
|---|---|---|---|---|
| GCN | Small | 0.15 | 0.37 | 0.43 |
| | Ogba | 0.59 | 1.07 | 1.41 |
| GAT | Small | 0.17 | 0.41 | 0.48 |
| | Ogba | 0.81 | 1.22 | 1.05 |
| GIN | Small | 0.07 | 0.26 | 0.20 |
| | Ogba | 0.59 | 0.98 | 1.50 |

We argue that this does not lead to scaling cost as the running time is reasonably small even for the largest OGB dataset with over 100k nodes.

**Optimal layer distribution.** To further analyze MetSelect, Figure 5 shows the proportion of the nodes that can be effectively classified by a GCN, GIN, GAT using its $l^*$ layer where $l^* \in \{0, 1, 2\}$. One can note that datasets like Cora, Citeseer, Pubmed, and ogbn-arxiv use higher-order graph structure for prediction, as noted by the high proportion of nodes with $l^* = 2$. On the other hand, datasets like Cornell, Wisconsin, and Texas benefit from a lower-order prediction (*i.e.*, layer $l^* = 0$). This shows that different datasets may have a different layer distribution that is ideal for predicting their node labels and MetSelect can adaptively find this underlying preference to certain representations in an automated manner.

### 7.5 How does MetSelect compare with layer aggregation methods?

Since layer aggregation methods (Xu et al., 2018b) make the predictions from an aggregated/pooled layer using LSTM, max, or concatenation operators, they cannot solve Problem 2. In other words, they cannot select one of the existing GNN representations as a personalized layer to predict the node class. Thus, we avoid such a comparison for fairness since unlike layer aggregation methods, we do not enhance existing models' representation power using non-linear activations.

For a fair comparison of the two methods, we create a setup where MetSelect also sees the set of pooled representations. Thus, we consider different layer aggregation strategies as multiple representations and select the best aggregation strategy among $\{\mathbf{h}^{(\mathrm{LSTM})}, \mathbf{h}^{(\mathrm{CAT})}, \mathbf{h}^{(\mathrm{MAX})}\}$. We find in Table 5 that MetSelect can be used to select personalized aggregation layers for improved performance than using any of the aggregation. This shows complementary gains of our method in addition to simple GNN layer selection.

## 8 Conclusion

Our work has shown that graph neural networks are further empowered by exploiting a personalized prediction layer for each node instead of using the same layer for all nodes. This challenges the common notion that the final layer representation of GNNs should fit all nodes. After formalizing the problem of finding the

node-optimal GNN layer, we proposed a novel method called MetSelect, to find the optimal layer for a given node in any given GNN. In particular, we use the layer that minimizes the distance of that node's representation from the class prototype in that layer. Results show that such a simple strategy can handle oversmoothing, boost performance, and robustness to attacks. We hope our work inspires further investigations into node-personalized training of GNNs, including regularization techniques for our method to learn more robust personalized representations. While it is extremely hard to prove that a particular layer is optimal for prediction, new benchmarks can be specifically designed for this task either synthetically or by sampling from real-world distributions. We also believe that theoretical investigations into how personalized decoding interacts with the training loss function can help gain useful insights into the convergence specificity of node classification of different nodes. Our work can also be extended to identifying personalized arbitrary motifs for each node for more fine-grained node classification. Automated layer selection can also inspire sparse architectures of the mixture of experts for graph neural networks (Shazeer et al., 2017).

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
