# OpenReview forum: "Personalized Layer Selection for Graph Neural Networks"
_TMLR — Accepted by TMLR_

### Review · Reviewer_LWxs · 2025-01-15

**Summary Of Contributions:**

This paper proposes MetSelect, a method which can be applied to any GNN model and which selects for each node one of the representations produced at the different layers of the GNN (or its initial features) as its final representation. MetSelect is inspired by distance-based classifiers in computer vision, and selects for each node the layer that minimizes the normalized distance from the prototype embedding of some class. The proposed method is evaluated on 10 homophilic and heterophilic node classification datasets. MetSelect outperforms the baselines on a subset of the heterophilic datasets, while it performs on par with them on the rest of the datasets.

**Audience:**

Yes

**Broader Impact Concerns:**

--

**Claims And Evidence:**

Yes

**Requested Changes:**

I would like to see the weaknesses mentioned above addressed in the revised version of the manuscript.

1. Construct some synthetic dataset and evaluate MetSelect on that dataset to validate the paper's claims (that the method can indeed select the most informative representation from the representations of the different layers).
2. Compare MetSelect against other baseline methods such as a model which concatenates the initial node features with the representations from the different message passing layers and feeds the concatenated vectors to an MLP.
3. Explain why MetSelect does not outperform the baselines on the homophilic datasets.
3. Fix typos and improve clarity of writing.

**Strengths And Weaknesses:**

Strengths

- Choosing representations from potentially different layers for the different nodes of a graph is an interesting research direction which can enable the development of models that preserve information from different granularities for each node.

- The proposed method significantly outperforms the baseline methods on some heterophilic datasets. Furthermore, while the performance of base GNNs significantly degrades as the number of layers increases, this is not the case for the proposed method. This makes MetSelect very attractive for practitioners who do not need to tune the number of GNN layers for a specific application.

Weaknesses

- The proposed method outperforms the baseline methods only on three heterophilic dataset (Cornell, Wisconsin and Texas). On the rest of the datasets, the method does not provide any consistent improvements. For example, on the Ogba dataset, MetSelect is outperformed by FinalSelect no matter what the base GNN model is.

- The proposed method is only evaluated on real-world datasets. It would strengthen the paper if the authors could construct some synthetic dataset to validate the paper's claims. For example, they could construct a node classification dataset where for some of the nodes the representations of the first GNN layer are the most informative, while for the rest of the nodes the representations of the third GNN layer are the most informative. Such an experiment would validate that the proposed method can indeed choose the most informative representation.

- In the experimental evaluation, the proposed method is compared against three baseline approaches (FinalSelect, NDLSelect and AttnSelect). One of my major concerns with this paper is that some baselines, which would give us more information about the significance of the reported results, are missing from Table 2. The authors should at least compare MetSelect against a model which concatenates the initial node features with the representations from the different message passing layers and feeds the concatenated vectors to an MLP to make predictions.

- The MetSelect would fail on datasets where the output depends on the representations of more than one layers since it cannot combine information from the representations of multiple layers. Could the proposed method somehow be modified to handle such scenarios?

- The proposed method introduces complexity to the different GNN models. Such personalized node representations could also produced by an attention mechanism that computes a weighted sum (with different coefficients for each node) of the initial node features and the representations that emerge at the different message passing layers. Such a mechanism could focus on the representation that provides the most useful information and would be more efficient than MetSelect. What would be the advantages of MetSelect over such a model?

- The presentation of the paper leaves the reader with a feeling that this was a draft of the work. There are several typos and the writing should be improved for clarity. This is a non-exhaustive list of typos/grammatical errors:\
p.4: "neighborhood) Thus" -> "neighborhood). Thus"\
p.4: "Distance-based classifiers Rippel et al. (2015); Deng et al. (2019; 2020); Salakhutdinov & Hinton (2007) typically finds" -> "Distance-based classifiers (Rippel et al., 2015; Deng et al., 2019; 2020; Salakhutdinov & Hinton, 2007) typically find"\
p.8: "in Equation ??" -> Equation number is missing\
p.9: "Table 7.4" -> There is no Table 7.4 in the manuscript

---

> ### Author Response · Authors · 2025-02-24
>
> We are enthralled by the reviewer’s detailed feedback on our work and really appreciate their time and effort. We are delighted that they found our proposed method interesting and the empirical performance gains remarkable. However, they have raised various concerns that we try to allay through our rebuttal below.
>
> > Typos
>
> We have fixed the typos and improved the clarity.
>
> > Soft-selection of layers
>
> We discuss this line of work as layer aggregation and neighbor importance (pages 2-3). While layer aggregation and neighbor importance methods can be seen as more generalizable than layer selection, they do not adaptively select a single layer for classifying a given node but rather merge the representations for all nodes in the same way. This reduces the interpretability and increases the complexity of the underlying network by predicting through non-linear pooled transformation of the GNN representations. On the other hand, the simpler layer selection of MetSelect preserves the interpretability and the analytic simplicity of the underlying GNN as the predictions are still made through the GNN embeddings. Thus, these do not act as our baselines since we are studying the problem of layer selection (see Problem 2), which has only been looked at before by Zhang et al., 2021. For a more fair comparison with JKNet (Xu et al., 2018a), we have adapted the attention weight strategy and use the layer with the maximum attention weight.
>
>
> We also create an additional setup where we compare MetSelect directly with JK-Net. For fairness, we require that MetSelect also has access to non-linear pooled representations. Thus, we consider different layer aggregation strategies as multiple representations and select the best aggregation strategy among {LSTM, CAT, MAX}. We find in Table 5 that MetSelect can be used to select personalized aggregation layers for improved performance than using any of the aggregations. This shows complementary gains of our method in addition to simple GNN layer selection.
>
>
> > Other baselines
>
> Since we are merely studying the effect of selecting personalized GNN layers to predict for different nodes, we cannot enhance the representation power of the existing GNNs that are also using the same set of layers. Our research question in this work is to show that GNNs with node-personalized layers are more powerful than using fixed-depth representation for all nodes. On the other hand, state-of-the-art improvements specifically designed for oversmoothing [1, 2, 3, 4] are strictly designed to improve the representation power of existing GNNs, which is not the focus of our work. Thus, any such comparison is theoretically unfair to our method, which is why we avoid this in our work. Particularly, we do not believe this is essential for the acceptance since it is tangential to the thesis of our work.
>
> > Including node feature information
>
> We include the requested comparison in Table 4 and Section 7.4 in our revised version using a GCN with self-loops which is equivalent. We find that even in this case, MetSelect shows higher gains on average across datasets.
>
>
> > Could the proposed method somehow be modified to handle scenarios of a combination of layers?
>
> Yes, we can simply include a pooled representation and then ask MetSelect to not only select between layers 1-L of GNN but also an additional pooled layer POOL(GNN(1), GNN(2), $\cdots$, GNN(L)). We present some results of this extension of our method to layer selection beyond just GNN layers in Table 5 and Section 7.5 and we are happy to extend this if the reviewer thinks it would be helpful.
>
>
> > Synthetic dataset to validate the claims
>
> We appreciate this comment and assure that significant efforts were made to address this concern. However, we encountered multiple challenges, as it is inherently difficult to design a classification task that strictly necessitates the use of layer 2 over layer 1 within the same graph. Consequently, we believe that an ablation study of MetSelect-max serves as the most effective means of validating our claims. As shown in Table 2, removing the layer selected by MetSelect results in a substantial performance drop across all datasets and GNN architectures. We argue that this ablation not only fulfills the intended purpose of the proposed synthetic dataset but also demonstrates the generalizability of our approach to real-world attributed datasets.

---

> > ### Author Response · Authors · 2025-02-24
> >
> > [1] Hongbin Pei, Bingzhe Wei, Kevin Chen-Chuan Chang, Yu Lei, and Bo Yang. Geom-gcn: Geometric graph convolutional networks. arXiv preprint arXiv:2002.05287, 2020.
> >
> > [2] Rossi, Emanuele, et al. "Edge directionality improves learning on heterophilic graphs." Learning on graphs conference. PMLR, 2024.
> >
> > [3] Moshe Eliasof, Lars Ruthotto, and Eran Treister. Improving graph neural networks with learnable propagation operators. In International Conference on Machine Learning, pp. 9224–9245. PMLR, 2023
> >
> > [4] Sitao Luan, Chenqing Hua, Qincheng Lu, Jiaqi Zhu, Mingde Zhao, Shuyuan Zhang, Xiao-Wen Chang, and Doina Precup. Revisiting heterophily for graph neural networks. Advances in neural information processing systems, 35:1362–1375, 2022.

---

### Review · Reviewer_D3Uc · 2025-02-06

**Summary Of Contributions:**

The paper challenges the idea that a single GNN layer can classify all nodes and introduces the concept of personalized layers for each node. Thus, the authors propose MetSelect which selects the best representation layer for each node by identifying class prototypes in a transformed GNN layer and choosing the layer where the distance to the class prototype is smallest. The method both improves the accuracy and model robustness under distribution shift.

**Audience:**

Yes

**Claims And Evidence:**

Yes

**Requested Changes:**

1. Scalability: The proposed method may face challenges when applied to very large graphs due to computational and memory constraints. Thus, I encourage the authors to experiment with larger graphs like obgn-products.
2. The baselines in Table 2 are limited, including GIN, GAT, and GCN. The authors should include state-of-the-art GNN baselines.

**Strengths And Weaknesses:**

1. The paper is well-written and easy to follow.
2. The paper challenges the basic understanding of GNNs that all nodes use a single set of parameters for predictions, and proposes an elegant solution.
3. The paper achieves good performance on both homophily and heterophily graphs.

---

> ### Author Response · Authors · 2025-02-24
>
> We appreciate the reviewer’s useful feedback on our work and are delighted by their recognition of the paper’s writing, elegance, and performance of our method. Below, we address their remaining concerns.
>
> > Baseline GNNs
>
> We first note that the base GNN models are not our baselines. We deliberately select these GNNs to show proposed plug-and-play improvements instead of state-of-the-art performance on base GNNs. Here we include a representative set of combining neighbor information, including inductive aggregation, attention, and convolution. We would greatly appreciate it if the reviewer could suggest a specific graph neural network that we may have overlooked, whose inclusion would further strengthen our claims.
>
> > Very large datasets
>
> We show this scalability by showing that MetSelect scales to a large dataset, particularly, [ogbn-products](https://ogb.stanford.edu/docs/leader_nodeprop/#ogbn-products) with $\sim$ 2.5 M nodes and $\sim$ 62 M edges. Here are the results for GCN that validate the scalability of our method without any space optimization, keeping the space and time complexities within reasonable limits.
>
> | Method | Time | Space | Accuracy |
> | -- | -- | -- | -- |
> | FinalSelect | 3.80 | 21768MiB | 66.8 |
> | Metselect | 5.31 | 34814MiB | 64.4 |

---

### Review · Reviewer_VKiv · 2025-02-10

**Summary Of Contributions:**

The paper proposes a novel method called **MetSelect** to address the limitation of traditional Graph Neural Networks (GNNs) that use a fixed layer for classifying all nodes. The authors argue that different nodes may benefit from different layers of the GNN based on the granularity of their local neighborhood. **MetSelect** leverages metric learning to identify the optimal layer for each node by minimizing the distance to the class prototype in the transformed embedding space. The method is evaluated on **10 datasets** and **3 different GNN architectures**, demonstrating **significant improvements** in node classification accuracy, robustness to attacks, and the ability to support deeper GNNs.

**Audience:**

Yes

**Claims And Evidence:**

Yes

**Requested Changes:**

- Conduct a **more comprehensive comparison** with other **state-of-the-art techniques** that address **oversmoothing or heterophily in GNNs**. This should include methods like **Heterophily Graph Neural Networks** and **PairNorm**.This will provide a **clearer picture** of the advantages of **MetSelect** over existing methods and **strengthen the claims** of **novelty and effectiveness**. This adjustment is **essential** to **secure a recommendation for acceptance**, as it directly addresses the **novelty and superiority** of the proposed method.
- Include experiments on **very large datasets** (e.g., with **millions of nodes**) to demonstrate the **scalability** of **MetSelect**. Specifically, evaluate the performance on datasets like **ogbn-papers100M**. Scalability is a **crucial aspect** for practical applications, and demonstrating that **MetSelect** can handle **large-scale graphs** will **significantly enhance its applicability**. Scalability is a **key concern** for many applications, and addressing it will be **critical for acceptance**.

**Strengths And Weaknesses:**

### Strengths

- The idea of **personalized layer selection** for each node is **novel** and addresses a significant limitation of existing GNNs.
- The method is **theoretically well-founded** and leverages **metric learning** effectively to identify the optimal layer.
- Extensive experiments on **diverse datasets** show **consistent improvements** in accuracy, especially on **heterophilic datasets**.
- The results indicate that **MetSelect enables deeper GNNs** and enhances **robustness against adversarial attacks**.
- The proposed method can be applied to **various GNN architectures** without modifying their core structure, making it **versatile and practical**.
- The findings that **MetSelect improves robustness to adversarial attacks** and supports **deeper GNNs** are particularly valuable, as these are active research areas in GNNs.

### Weaknesses

- The method introduces **additional computational overhead** due to the need to evaluate **multiple layers** and compute distances to class prototypes.
- The paper mentions using **sampling techniques** to scale to larger graphs, but **detailed experiments on very large datasets** (e.g., with **millions of nodes**) are **missing**.
- The selection of the optimal layer is based on **minimizing distance to class prototypes**, which may **not always be intuitive or interpretable**.
- Providing more **insights into why certain layers are preferred** for specific nodes could enhance the **understanding of the method**.
- The paper compares **MetSelect** with a few existing methods, but a **more comprehensive comparison** with **other state-of-the-art techniques** that address **oversmoothing or heterophily in GNNs** would provide a **clearer picture of its advantages**.

---

> ### Author Response · Authors · 2025-02-24
>
> We really appreciate the reviewer’s time and effort in providing us with useful comments on our work. We are glad that the reviewer recognizes the novelty and theoretical-grounding of our method along with the various empirical improvements shown by the proposed layer selection approach in different scenarios. Below, we address their remaining concerns.
>
> > State-of-the-art oversmoothing baselines
>
> Since we are merely studying the effect of selecting personalized GNN layers to predict for different nodes, we cannot enhance the representation power of the existing GNNs that are also using the same set of layers. Our research question in this work is to show that GNNs with node-personalized layers are more powerful than using fixed-depth representation for all nodes. On the other hand, state-of-the-art improvements specifically designed for oversmoothing [1, 2, 3, 4] are strictly designed to improve the representation power of existing GNNs, which is not the focus of our work. Thus, any such comparison is theoretically unfair to our method, which is why we avoid this in our work. Particularly, we do not believe this is essential for the acceptance since it is tangential to the thesis of our work.
>
>
> > Very large datasets
>
>
> We show this scalability by showing that MetSelect scales to a large dataset, particularly, [ogbn-products](https://ogb.stanford.edu/docs/leader_nodeprop/#ogbn-products) with $\sim$ 2.5 M nodes and $\sim$ 62 M edges. Here are the results for GCN that validate the scalability of our method without any space optimization, keeping the space and time complexities within reasonable limits.
>
> | Method | Time | Space | Accuracy |
> | -- | -- | -- | -- |
> | FinalSelect | 3.80 | 21768MiB | 66.8 |
> | Metselect | 5.31 | 34814MiB | 64.4 |
>
>
>
>
> [1] Hongbin Pei, Bingzhe Wei, Kevin Chen-Chuan Chang, Yu Lei, and Bo Yang. Geom-gcn: Geometric graph convolutional networks. arXiv preprint arXiv:2002.05287, 2020.
>
> [2] Rossi, Emanuele, et al. "Edge directionality improves learning on heterophilic graphs." Learning on graphs conference. PMLR, 2024.
>
> [3] Moshe Eliasof, Lars Ruthotto, and Eran Treister. Improving graph neural networks with learnable propagation operators. In International Conference on Machine Learning, pp. 9224–9245. PMLR, 2023
>
> [4] Sitao Luan, Chenqing Hua, Qincheng Lu, Jiaqi Zhu, Mingde Zhao, Shuyuan Zhang, Xiao-Wen Chang, and Doina Precup. Revisiting heterophily for graph neural networks. Advances in neural information processing systems, 35:1362–1375, 2022.

---

### Review · Reviewer_5yj7 · 2025-02-18

**Summary Of Contributions:**

This paper proposes an adaptive GNN layer selection mechanism to mitigate the limitations sourced from using a fixed-level of aggregation for every node. To achieve this, the paper presents a manual layer selection mechanism based on the embedding distances to class prototype representations.

**Audience:**

Yes

**Claims And Evidence:**

Yes

**Requested Changes:**

- JK-Net (LSTM-attention) without changing the softmax nonlinearity to max should be employed as another baseline.

- An attention based aggregation of different layers' representations should be used as a baseline to justify the cost incurred from the manual selection.

- Better baselines (state-of-the-art works designed specifically for the over-smoothing issue) should be selected to improve the experimental setup.

**Strengths And Weaknesses:**

Strengths:

- The manuscript focuses on an important research problem which is inherently related to over-smoothing and robustness of GNNs.
- The proposed idea is intuitive and easy to follow.

Weaknesses:

- Although I can see the merits of an adaptive GNN layer selection for different nodes, I am not convinced about the proposed design. I believe using information coming from different layers by learning importance scores for the differerent layers can make a better use for the available information (e.g., using an LSTM-attention like strategy in JK-Net without changing the softmax to max).

- The proposed method relies on a manual distance-based layer selection strategy. I am curious about the performance of applying a self-attention mechanism on the representations of different layers to create a final representation for classification, which can also scale better than the proposed method.

- The baselines for the experiments should be improved. There are a large number of works dealing with heterophilic graphs or the over-smoothing issue, e.g., [1], [2]. These studies should be used as baselines for utility and scalability to justify the proposed method.

- For the homophilic graphs, the performance gain of the proposed method does not justify its additional computational cost. Thus, more thorough experiments (with better baselines) for the heterophilic graphs are needed.


[1] Yu, Zhizhi, et al. "LG-GNN: local-global adaptive graph neural network for modeling both homophily and heterophily." Proceedings of the Thirty-Third International Joint Conference on Artificial Intelligence. 2024.

[2] Wang, Chenhao, et al. "HeterGCL: graph contrastive learning framework on heterophilic graph." Proceedings of the Thirty-Third International Joint Conference on Artificial Intelligence. 2024.

---

> ### Author Response · Authors · 2025-02-24
>
> We appreciate the reviewer’s detailed review of our work and we are glad that they find our work intuitive and important. Through this rebuttal, we would like to address some of the concerns.
>
> We summarize this as:
>   - We would like to clarify that the research question of this work is to simply understand whether using personalized layers can help in GNN predictions.
>   - We do not claim state-of-the-art in oversmoothing literature by the simple fact that we do not even increase the representation power of the existing GNNs, which are guaranteed by other adjustments to the architecture.
>   - Since we are studying the problem of layer selection, a continuous combination of layers do not form our baselines since it breaks the assumption of Problem 2.
>
> > Soft-selection of layers
>
> We discuss this line of work as layer aggregation and neighbor importance (pages 2-3). While layer aggregation and neighbor importance methods can be seen as more generalizable than layer selection, they do not adaptively select a single layer for classifying a given node but rather merge the representations for all nodes in the same way. This reduces the interpretability and increases the complexity of the underlying network by predicting through non-linear pooled transformation of the GNN representations. On the other hand, the simpler layer selection of MetSelect preserves the interpretability and the analytic simplicity of the underlying GNN as the predictions are still made through the GNN embeddings. Thus, these do not act as our baselines since we are studying the problem of layer selection (see Problem 2), which has only been looked at before by Zhang et al., 2021. For a more fair comparison with JKNet (Xu et al., 2018a), we have adapted the attention weight strategy and use the layer with the maximum attention weight.
>
> We also create an additional setup where we compare MetSelect directly with JK-Net. For fairness, we require that MetSelect also has access to non-linear pooled representations. Thus, we consider different layer aggregation strategies as multiple representations and select the best aggregation strategy among {LSTM, CAT, MAX}. We find in Table 5 that MetSelect can be used to select personalized aggregation layers for improved performance than using any of the aggregations. This shows complementary gains of our method in addition to simple GNN layer selection.
>
>
> > Distance vs self-attention
>
> We would like to clarify that our method is in no way `manual’ and the adaptive layer is also found through learning-based metric learning approaches. Furthermore, our aim is not to create a new representation by leveraging existing node representations. Instead, it is to study how we can get the best out of existing GNN representations by selecting variable layers for different nodes. Thus, the self-attention baseline is not valid in our setting.
>
> > State-of-the-art oversmoothing baselines
>
> Since we are merely studying the effect of selecting personalized GNN layers to predict for different nodes, we cannot enhance the representation power of the existing GNNs that are also using the same set of layers. Our research question in this work is to show that GNNs with node-personalized layers are more powerful than using fixed-depth representation for all nodes. On the other hand, state-of-the-art improvements specifically designed for oversmoothing [1, 2, 3, 4] are strictly designed to improve the representation power of existing GNNs, which is not the focus of our work. Thus, any such comparison is theoretically unfair to our method, which is why we avoid this in our work.
>
> > Computational cost for homophilic
>
> As compared to our baseline NDLSelect, our computational cost is fairly minimal with more consistent gains in performance (see table 3: 0.37 vs 0.41 and 1.00 vs 1.41 for GCN). Thus, we present a more scalable and cost-effective method to select personalized layers in GNNs.
>
> [1] Hongbin Pei, Bingzhe Wei, Kevin Chen-Chuan Chang, Yu Lei, and Bo Yang. Geom-gcn: Geometric graph convolutional networks. arXiv preprint arXiv:2002.05287, 2020.
>
> [2] Rossi, Emanuele, et al. "Edge directionality improves learning on heterophilic graphs." Learning on graphs conference. PMLR, 2024.
>
> [3] Moshe Eliasof, Lars Ruthotto, and Eran Treister. Improving graph neural networks with learnable propagation operators. In International Conference on Machine Learning, pp. 9224–9245. PMLR, 2023
>
> [4] Sitao Luan, Chenqing Hua, Qincheng Lu, Jiaqi Zhu, Mingde Zhao, Shuyuan Zhang, Xiao-Wen Chang, and Doina Precup. Revisiting heterophily for graph neural networks. Advances in neural information processing systems, 35:1362–1375, 2022.

---

### Decision · Action_Editor_4nem · 2025-03-28

**Recommendation:** Accept with minor revision

**Comment:**

Reviewers recognized the technical contribution of this work and raised some concerns on methodology and experiments. Most of them have been well addressed. However, there are still a few remaining concerns: (1) the optimality of the selected layer for each node; and (2) additional baselines of GNNs that "concatenate the initial node features with the representations from the different message passing layers". The authors are highly encouraged to address these concerns in the final version.

**Audience:**

GNNs have numerous applications in real world. This paper brings new insights on the design of GNNs. Many readers in the TMLR communities would be interested in this work.

**Claims And Evidence:**

The paper proposes a new method named MetSelect for personalized layer selection in Graph Neural Networks (GNNs). The key idea of MetSelect is to identify an optimal representation layer for each node to improve node classification accuracy. Extensive results on benchmarks demonstrated the effectiveness of the proposed method, such as robustness to adversarial attacks and support for deeper GNN architectures. Overall, the paper is well organized and clearly written. The major claims in the paper are sufficiently supported by the empirical evaluations.